# In Vitro and In Vivo Neuroprotective Effects of Stellettin B Through Anti-Apoptosis and the Nrf2/HO-1 Pathway

**DOI:** 10.3390/md17060315

**Published:** 2019-05-29

**Authors:** Chien-Wei Feng, Nan-Fu Chen, Zhi-Hong Wen, Wen-Ya Yang, Hsiao-Mei Kuo, Ping-Jyun Sung, Jui-Hsin Su, Shu-Yu Cheng, Wu-Fu Chen

**Affiliations:** 1National Museum of Marine Biology and Aquarium, Pingtung 944, Taiwan; qscjuejuejue@gmail.com (C.-W.F.); pjsung@nmmba.gov.tw (P.-J.S.); x2219@nmmba.gov.tw (J.-H.S.); 2Department of Marine Biotechnology and Resources, National Sun Yat-sen University, Kaohsiung 804, Taiwan; wzh@mail.nsysu.edu.tw (Z.-H.W.); s8222889@gmail.com (W.-Y.Y.); hsiaomeikuo@gmail.com (H.-M.K.); 3Division of Neurosurgery, Department of Surgery, Kaohsiung Armed Forces General Hospital, Kaohsiung 802, Taiwan; chen06688@gmail.com; 4Department of Neurological Surgery, Tri-Service General Hospital, National Defense Medical Center, Taipei 114, Taiwan; 5Doctoral Degree Program in Marine Biotechnology, National Sun Yat-sen University, Kaohsiung 804, Taiwan; joygetit@gmail.com; 6Center for Neuroscience, National Sun Yat-Sen University, Kaohsiung 804, Taiwan; 7Graduate Institute of Marine Biology, National Dong Hwa University, Pingtung 944, Taiwan; 8Doctoral Degree Program in Marine Biotechnology, Academia Sinica, Nankang, Taipei 115, Taiwan; 9Department of Neurosurgery, Kaohsiung Chang Gung Memorial Hospital and Chang Gung University College of Medicine, Kaohsiung 833, Taiwan

**Keywords:** sponge-derived compound, Parkinson’s disease, oxidative stress, apoptosis, Nrf2, HO-1

## Abstract

Pharmaceutical agents for halting the progression of Parkinson’s disease (PD) are lacking. The current available medications only relieve clinical symptoms and may cause severe side effects. Therefore, there is an urgent need for novel drug candidates for PD. In this study, we demonstrated the neuroprotective activity of stellettin B (SB), a compound isolated from marine sponges. We showed that SB could significantly protect SH-SY5Y cells against 6-OHDA-induced cellular damage by inhibiting cell apoptosis and oxidative stress through PI3K/Akt, MAPK, caspase cascade modulation and Nrf2/HO-1 cascade modulation, respectively. In addition, an in vivo study showed that SB reversed 6-OHDA-induced a locomotor deficit in a zebrafish model of PD. The potential for developing SB as a candidate drug for PD treatment is discussed.

## 1. Introduction

Parkinson’s disease (PD) is the second most common neurodegenerative disease after Alzheimer’s disease, with the global PD prevalence estimated to be 6.3 million [1]. To date, therapies have only succeeded in relieving clinical symptoms, but not arresting disease progression, and therefore, preventative or therapeutic agents to combat PD are urgently needed [2,3]. Development of novel PD drugs requires an understanding of PD etiology, and among several factors contributing to the development of PD, oxidative stress is possibly the most important [4]. One of the most widely utilized pathway for counteracting intracellular oxidative stress, via the phase II enzyme [5,6], is the antioxidant response element (ARE) [4], which was originally discovered in 1990 in the promoter region of glutathione S-transferase (GSTA2) [7]. Studies have found that activated nuclear factor erythroid 2-related factor 2 (Nrf2) plays a crucial role in triggering expression of the ARE pathway components [8,9] through translocation to the nucleus and by initiating downstream signaling by binding to the ARE domain [10]. The neuroprotective effect of Nrf2 modulation in PD has been demonstrated [11] via its downstream expression products, including heme-oxygenase 1 (HO-1), superoxide dismutase 1 (SOD-1), and NAD(P)H dehydrogenase quinone 1 [12,13].

HO-1 is localized to the endoplasmic reticulum and acts as a restrictive enzyme in heme metabolism. It catalyzes heme oxidation to produce ferrous ions (Fe^2+^), carbon monoxide, and bilirubin [14]. Serum levels of bilirubin have been shown in clinical trials to be significantly reduced and accompanied by a massive increase in the production of free radicals in PD patients [15]. Complementing this finding, others have shown that bilirubin exhibits anti-apoptotic activity [16,17,18]. Together, these results suggest that HO-1 may exert significant therapeutic effects with respect to PD.

SOD-1 plays a key role in countering oxidative stress and catalyzing the conversion of superoxide anion radicals to hydrogen peroxide (H_2_O_2_) and oxygen (O_2_) through a disproportionation reaction. The bioactivity of SOD-1 in red blood cells has been demonstrated in clinical trials to be substantially reduced in PD patients [19], highlighting the vital role of SOD-1 in PD [20,21]. In addition, increasing evidence supports the notion that oxidative stress and apoptosis are closely linked physiological processes that are implicated in the pathophysiology of PD [22].

Apoptosis is affected by numerous pathways, included PI3K/Akt, MAPK/P38, and the caspase cascade. Numerous pathways including PI3K/Akt, MAPK/P38, and caspases modulate apoptosis. The roles of the PI3K/Akt pathway in regulating neuronal cell survival, proliferation, and differentiation have been widely investigated in PD [23,24]. On the other hand, MAPK/P38 modulates stress responses and apoptosis in dopaminergic neurons [25]. Moreover, previous studies indicate that activation of PI3K/Akt and inhibition of MAPK/P38 block apoptosis and promote dopaminergic cell survival [26,27]. Thus, the modulatory effect of candidate compounds on the PI3K/Akt or MAPK/P38 pathways may be import in exerting neuroprotective activity. The ocean serves as a potential biological resource for novel PD drug discovery due to its biological diversity. Also, marine-derived drugs may solve the current drug shortage issues [28].

Marine organism-derived compounds may have potential as novel anti-PD drug candidates [29], similar to our previous findings for various diseases [30,31,32]. Stellettin B (SB) is an isomalabaricane triterpenoid isolated from the marine sponge *Jaspis stellifera*, which was first identified by Boyd and Paull in 1995. The chemical structure of SB was shown (Figure 1). SB exhibited antitumor activity against SF295 glioblastoma cells from a mouse lymphocytic leukemia at a concentration of 1 μΜ [33,34]. Subsequent studies implicated the PI3K/Akt pathway in the antitumor property of SB and in PD [33,35,36,37,38]. Our recent study demonstrated that SB exhibits anti-invasion and anti-angiogenesis effects likely through the Akt/Girdin pathway in glioblastoma, a common and aggressive malignant primary cancer of the central nervous system [39]. However, the neuroprotective function of SB has not been investigated.

In this study, we determined the anti-apoptotic and anti-oxidative stress effects of SB in 6-OHDA-induced apoptosis in the SH-SY5Y cell line using Western blotting, TUNEL, and CellROX^®^. In addition, we also confirmed the neuroprotective activity of SB in a zebrafish model of PD. We intend to extrapolate these findings into clinical studies with the aim of helping patients diagnosed with PD.

## 2. Results

### 2.1. Protective Effect of SB on 6-OHDA-Induced Cell Death in SH-SY5Y Cells

The neuroprotective effect of SB on 6-OHDA-induced SH-SY5Y cell damage was determined using an alamarBlue assay. SH-SY5Y cells were pretreated with 0.1, 1, 10, or 100 nM SB for 1 h before incubation with 20 μM 6-OHDA for 16 h. All SB concentrations tested significantly (*p* < 0.05) protected SH-SY5Y cells against 6-OHDA-induced cell damage (Figure 2A), and therefore, 0.1 nM was used in all subsequent experiments. Hoechst 33342 staining was used to validate cell apoptosis. Treatment with 20 μM 6-OHDA for 8 h significantly condensed the chromatin, representing apoptotic cells. However, this effect was significantly inhibited by pretreatment with 0.1 nM SB (*p* < 0.05), and 0.1 nM SB alone did not cause massive cell death (Figure 2B). When quantified, treatment with 20 μM 6-OHDA resulted in 30% cell death, which was reduced to 5% by pretreatment with 0.1 nM SB for 1 h (Figure 2C).

### 2.2. The Anti-Apoptotic Effect of SB on 6-OHDA-Induced Cytotoxicity

Apoptosis was quantified using TUNEL staining, wherein the enzyme terminal deoxynucleotide transferase attaches deoxynucleotides to the 3′-hydroxyl terminus of DNA breaks that are formed when DNA fragmentation occurs in the last phase of apoptosis. Incubation with 20 μM 6-OHDA for 8 h clearly increased TUNEL staining compared with the control group, and administration of 0.1 nM SB significantly reduced the number of TUNEL-positive cells (*p* < 0.05). SB alone did not produce a significant change in the number of TUNEL-positive cells (Figure 3A). Quantification showed that 6-OHDA increased apoptotic cell numbers from 2.3% to 40% of the total, whereas pretreatment with 0.1 nM SB significantly attenuated 6-OHDA-induced apoptosis of SH-SY5Y cells (Figure 3B) (*p* < 0.05). We then examined caspase-3 protein expression by Western blotting to further confirm the relationship between SB and its anti-apoptotic activity. Exposure of 20 μM 6-OHDA for 8 h significantly increased the expression of activated caspase-3, whereas SB significantly blocked its activation (Figure 3C,D) (*p* < 0.05). Uncropped Western blots of caspase-3 and β-actin were shown in Appendix A.

### 2.3. Effect of SB on Phosphorylation of Extracellular Signal-Regulated Kinases (Phospho-ERK), Protein Kinase B (Phospho-Akt), and P38 (Phospho-P38) in 6-OHDA-Treated SH-SY5Y Cells

The levels of phospho-ERK, phospho-Akt, and phospho-P38 proteins, which play an important role in neuronal cell survival, were analyzed by Western blotting. Treatment of SH-SY5Y cells with 6-OHDA led to a notable downregulation of phospho-ERK between 15 and 120 min, but this was significantly reversed by pretreatment with 0.1 nM SB for 60 min (*p* < 0.05); however, incubation with SB alone did not affect phospho-ERK levels (Figure 4). Uncropped Western blots of p-ERK and ERK were shown in Appendix A. Similarly, treatment with 6-OHDA downregulated phospho-Akt between 60 and 120 min, which was significantly reversed by pretreatment with 0.1 nM SB (*p* < 0.05). As well, treatment with 0.1 nM SB alone did not affect phospho-Akt levels. Uncropped Western blots of p-Akt and Akt were shown in Appendix A.

While ERK and Akt signaling promotes cell survival, phosphorylation of P38 MAPK induces apoptosis. By Western blotting, we found that treatment with 20 μM 6-OHDA significantly upregulated phospho-P38 levels between 15 and 120 min (Figure 4) (*p* < 0.05). Pretreatment with 0.1 nM SB attenuated this 6-OHDA-induced upregulation of phospho-P38 between 30 and 60 min. Uncropped Western blots of p-P38 and P38 were shown in Appendix A.

### 2.4. Effect of SB on 6-OHDA-Induced Oxidative Stress in SH-SY5Y Cells

Determination of the effect of SB on 6-OHDA-induced oxidative stress in SH-SY5Y cells showed that 6-OHDA treatment significantly increased reactive oxygen species (ROS)-positive cell numbers from 3 ± 1.5% to 38.6 ± 5.9% of the total, whereas pretreatment with 0.1 nM SB significantly attenuated this increase (12.3 ± 1.5%; Figure 5A) (*p* < 0.05). Furthermore, 0.1 nM SB treatment alone had no effect on the generation of cellular oxidative stress. Quantification showed that 0.1 nM SB significantly inhibited 6-OHDA-induced upregulation of ROS-positive cell numbers (Figure 5B) (*p* < 0.05). In addition, we also examined the effect of SB on anti-oxidative stress using superoxide dismutase (SOD) activity. Treatment with 6-OHDA decreased cellular SOD activity from 100 ± 0.9% to 63.5 ± 4.4%, and pretreatment with 0.1 nM SB significantly reversed 6-OHDA-induced downregulation to 94.1 ± 9.7% (*p* < 0.05). More importantly, 0.1 nM SB alone also increased SOD activity by almost 40% (Figure 5C).

### 2.5. Effect of SB on the Nrf2-ARE Signaling-Related Pathway in 6-OHDA-Induced SH-SY5Y Cells

Examination of the Nrf2-ARE signaling pathway and induction of downstream molecules, such as the antioxidants, HO-1, and glutathione, showed that treatment with SB (0.1 nM) alone for 1, 3, and 6 h significantly increased Nrf2 protein expression with lamin B1 as the nuclear internal control, but cytoplasmic Nrf2 showed no difference (Figure 6A) (*p* < 0.05). Uncropped Western blots of Nuclear-Nrf2 and Cytoplasm-Nrf2 were shown in Appendix A. Treatment with SB for 24 h and from 0.1 to 10 nM significantly enhanced HO-1 expression (Figure 6B) (*p* < 0.05), whereas 0.1 nM SB treatment significantly upregulated HO-1 expression between 16 and 24 h (Figure 6C) (*p* < 0.05). Uncropped Western blots of HO-1 and β-actin were shown in Appendix A.

We then tested the effects of SB on the Nrf2–ARE pathway upon 6-OHDA-treatment. After a 3 h exposure to 6-OHDA, expression of Nrf2 increased significantly (*p* < 0.05). Pretreatment with 0.1 nM SB for 1 h further significantly increased Nrf2 expression (Figure 7A) (*p* < 0.05). Uncropped Western blots of Nuclear-Nrf2, Cytoplasm-Nrf2 and Lamin b1 were shown in Appendix A. Treatment with 0.1 nM SB alone also significantly increased Nrf2 expression (*p* < 0.05). Similarly, pretreatment with 0.1 nM SB for 1 h also enhanced 6-OHDA-induced upregulation of HO-1 expression after exposure to 6-OHDA for 24 h. Finally, 0.1 nM SB treatment alone also significantly upregulated HO-1 expression (Figure 7B) (*p* < 0.05). Uncropped Western blots of HO-1 and β-actin were shown in Appendix A.

### 2.6. Effect of PI-3K and HO-1 Inhibitors on Modulation of the SB Neuroprotective Effect in 6-OHDA-Induced SH-SY5Y Cells

To examine the effect of SB on the PI3K and Nrf2–ARE–HO-1 pathways, we utilized the PI3K inhibitor LY294002 and the HO-1 inhibitor, zinc protoporphyrin (ZnPP) in order to further confirm the therapeutic mechanism of action of SB. As stated earlier, 0.1 nM SB conferred significant neuroprotection to 6-OHDA-treated cells, whereas LY294002 (1 and 10 μM) significantly attenuated SB-induced neuroprotection after 6-OHDA challenge (Figure 8A) (*p* < 0.05). In addition, we also used ZnPP to assess SB’s therapeutic mechanism through HO-1. As shown earlier in this study, 0.1 nM SB induced significant neuroprotection in 6-OHDA-treated cells. The HO-1 inhibitor, ZnPP (10 μM), significantly attenuated SB-induced neuroprotection after the 6-OHDA challenge (Figure 8B) (*p* < 0.05).

### 2.7. The Protective Effect of SB on 6-OHDA-Induced Locomotor Deficiency and Tyrosine Hydroxylase (TH) Attenuation in Zebrafish

The therapeutic effect of SB on 6-OHDA-induced cellular toxicity was extrapolated to an in vivo zebrafish PD model that was established previously by our group [40]. Exposure to 250 μΜ 6-OHDA (2 to 5 days post-fertilization (dpf)) reduced the total swimming distance from 955.9 ± 109.4 to 40.7 ± 21.2 mm. However, pretreatment with 1 nM SB (9 h post-fertilization (hpf) to 5 dpf) significantly reversed the total swimming distance from 40.7 ± 21.2 to 861 ± 171.6 mm (Figure 9A) (*p* < 0.05). The number of dopaminergic neurons was determined by examining TH protein expression in 6-OHDA-treated zebrafish by Western blotting. We found that treatment with 250 μΜ 6-OHDA between 2 and 5 dpf significantly decreased TH protein expression from 100 ± 7.6% to 70.9 ± 5.4% (*p* < 0.05). However, pretreatment with 1 nM SB (9 hpf to 5 dpf) significantly reversed this effect from 70.9 ± 5.4% to 91.1 ± 6.2% (*p* < 0.05). Treatment with 1 nM SB (9 hpf to 5 dpf) alone did not affect TH protein expression (Figure 9B). Uncropped Western blots of TH and β-actin were shown in Appendix A.

## 3. Discussion

Current mainstream treatment options for PD include L-3,4-dihydroxyphenylalanine monoamine oxidase-B (MAO-B) inhibitors, Catechol-O-methyl transferase (COMT) inhibitors, amantadine, and anticholinergic agents. Although these therapies can relieve clinical symptoms, they do not slow down disease progression [41]. Findings from clinical studies point to PD drugs that focus on etiology as potentially being effective in arresting disease progression [42,43,44]. Therefore, in this study, we investigated whether the marine sponge-derived bioactive molecule, SB, can protect SH-SY5Y cells from 6-OHDA-induced cell death. SB exerts its neuroprotective activity via both anti-apoptotic and anti-oxidative stress pathways, and in this regard, PI3K/Akt, ERK, and P38 have been widely discussed as relevant anti-apoptotic pathways in previous PD-related research. The PI3K/Akt pathways play a crucial role in dopaminergic cell survival in PD [27,45], and, therefore, molecules such as rifampicin and resveratrol that act via PI3K/Akt pathways have been investigated as potential therapeutic agents in in vitro models of PD [46,47,48]. Wu et al. reported that treatment with 10 μM rifampicin could significantly inhibit rotenone-induced cell apoptosis via activation of the PI3K/Akt pathway [48], and systemic treatment with rifampicin was neuroprotective in a murine PD model [49]. Similarly, Lin et al. demonstrated that resveratrol can significantly inhibit rotenone-induced SH-SY5Y cell apoptosis via the PI3K/Akt pathway [50]. Moreover, resveratrol was neuroprotective in a rat PD model, in which the compound significantly reversed 6-OHDA-induced locomotor deficits and TH decline [51].

We found that SB activated the PI3K/Akt pathway, and compared with epigallocatechin gallate (EGCG) (10 μM) and resveratrol (10 mM), even a 0.1 nM concentration of SB was sufficient to induce a potent anti-apoptotic effect (Figure 4), a property that benefits clinical drug development. With respect to apoptosis of dopaminergic cells in PD [27,52,53], ERK, and P38 activators such as sulfuretin and chrysotoxine have shown efficacy in in vitro models of PD, by blocking apoptosis. Pariyar et al. showed that sulfuretin protects neuronal cells against MPP^+^-induced damage through the activation of ERK and its downstream anti-apoptosis effector, Bcl-2, in SH-SY5Y cells [54]; however, sulfuretin has not been studied in an in vivo PD model. In addition, sulfuretin protected dopaminergic neurons from amyloid beta neurotoxicity in an in vitro model of Alzheimer’s disease [55]. By partially or totally inhibiting pro-apoptotic signals induced by MPP^+^, chrysotoxine has been shown to exert a neuroprotective activity [56]. Thus, sulfuretin and chrysotoxine exert their neuroprotective activities via p-ERK activation and P38 inhibition similar to that shown by SB; however, these functional properties of sulfuretin and chrysotoxine were not demonstrated in an in vivo model. The present study showed that SB can exert neuroprotective activity by enhancing p-ERK and attenuating P38 phosphorylation in an in vitro model of 6-OHDA-mediated neurotoxicity and this was further confirmed in an in vivo zebrafish PD model.

Previous studies have revealed that PI3K/Akt and ERK can both possibly act on the downstream Nrf2/ARE pathway. Because of the important roles of oxidative stress and apoptosis in PD progression, the Nrf2/HO-1 pathway is considered a promising PD therapeutic target [57,58]. Potential candidate drugs including puerarin and deprenyl (phenyl-isopropyl-methyl-propargylamine) exert their protective effects via the Nrf2/HO-1 pathway. Puerarin is an isoflavone found in several plants that exerted neuroprotective activity in a rat PD model by upregulating BDNF expression and inhibited oxidative stress damage by activating the Nrf2/ARE signaling pathway in the substantia nigra (SN) [59]. Deprenyl, an MAO-B inhibitor used in the treatment of PD achieves its neuroprotective effect by also activating the Nrf2/HO-1 pathway, as demonstrated in SH-SY5Y cells [60,61]. Deprenyl also significantly rescued MPTP-induced locomotor deficits and increased TH expression in the SN of an in vivo rat PD model [62]. Our results indicate that, similar to deprenyl and puerarin, SB may amplify the effects of the Nrf2/HO-1 pathway. However, unlike these studies, we also investigated the effect of SB on SOD activity and also used ZnPP to show that SB’s neuroprotective activity was partially mediated by HO-1.

Some research also showed a similar result to the zebrafish model as our study. Xu et al. showed that Loganin, one of the best-known iridoid glycosides, exerts a neuroprotective effect in an MPTP-mouse model of PD through anti-inflammation, autophagy attenuation, and anti-apoptotic activities [63]. As well, loganin can rescue MPTP-induced locomotor deficiencies and the decline of dopaminergic neurons in zebrafish larvae [64]. Furthermore, treatment of zebrafish with rotenone for 6 h at 48 hpf induces mitochondrial dysfunction and loss of TH-positive cells [65], and using previously described methods, Chong et al. induced PD-like symptoms [66]. Zhang et al. showed that berberine can rescue 6-OHDA-induced (from 2 to 4 dpf) loss of dopaminergic neurons and the deficiency in movement behavior in zebrafish [67]. In our previous study, we established a PD model in zebrafish larvae and demonstrated that vitamin E, minocycline, and sinemet can rescue 6-OHDA-induced locomotor deficiency [40]. In the present study, we have shown that the administration of SB followed by 6-OHDA treatment leads to an improvement of locomotor activity and recovery of TH protein expression in zebrafish brain tissue.

Currently, PD drugs focus on improving clinical symptoms using alternative dopamine agonists or on alleviating the metabolism of existing endogenous dopamine. However, no compound can reverse or stop PD progression. To expand the range of drugs available to tackle PD, we searched for and identified marine sponge-derived SB, which was hypothesized not to possess neuroprotective properties based on the previous literature. Earlier studies focused on SB’s antitumor effects in human non-small cell lung cancer and glioblastoma [33,37,39]. To our excitement, we demonstrated its neuroprotective activity in a PD model, albeit at a lower dosage. Interestingly, this phenomenon is also observed in the clinical painkiller ziconotide, which is a novel non-opioid analgesic drug whose peptide was isolated from the venom of the fish-eating marine snail, *Conus magus*, and possesses a strong analgesic effect at the appropriate dosage but is toxic in higher doses. In addition, we have shown previously that a coral-derived compound, 11-dehydrosinualriolide, also exerts a neuroprotective effect in a PD model [68,69] and has also been investigated for its antitumor effect [70,71]. We will continue to investigate the effect of SB treatment in zebrafish and rat models of PD, and take it to a clinical trial, as we move toward our goal of going from bench to bedside.

## 4. Materials and Methods 

### 4.1. Ethical Approval

The National Sun Yat-sen University Animal Care and Use Committee approved the care of animals and experiments. Our procedure followed the Guiding Principles in the Care and Use of Animals of the American Physiology Society. All studies involving animals are reported in accordance with the ARRIVE guidelines for reporting experiments involving animals. Each zebrafish was used only once during the study.

### 4.2. SB Preparation

SB was isolated from the marine sponge *Jaspis stellifera*, which was kindly provided by Ping-Jyun Sung and Jui-Hsin Su (National Museum of Marine Biology & Aquarium, Taiwan) and is shown in Figure 1. SB stock (100 mM) was stored in DMSO at −20 °C. Stellettin B was obtained from the sponge, *Jaspis stellifera* (1.2 kg, wet wt). The freeze-dried sponge (250g) was thoroughly extracted with methanol. Then, the methanol extract was subjected to further partition between H_2_O and CH_2_Cl_2_. The CH_2_Cl_2_ layer was separated over a silica gel by column chromatography to obtain stellettin B (200 mg).

### 4.3. Cell Culture

SH-SY5Y cells were obtained from American Type Culture Collection (Manassas, VA, USA) and cultured at 37 °C and 5% CO_2_ with DMEM medium containing 2 mM L-glutamine, 20 U/mL penicillin–streptomycin, 5 mg/mL gentamicin, and 15% (v/v) fetal bovine serum. Cell numbers seeded for each assay were 2 × 10^4^ cells/well in 96-well microplates (Corning, NY, YSA) for Hoechst staining, neuroprotection assay, and CellROX^®^ assay; 1.8 × 10^6^ cells/well in 6-well culture plate containing coverslips (24 × 24 mm) for TUNEL staining; and 1 × 10^6^ cells/dish in 10 cm dish for Western blot analysis.

### 4.4. SH-SY5Y Cell Neuroprotection Assay

Neuroprotection assays were performed using alamarBlue^®^ reagent (Invitrogen, MA, USA). Briefly, SH-SY5Y cells were pretreated with SB for 1 h before adding 20 μM 6-hydroxydopamine (6-OHDA; Sigma, St. Louis, MO, USA) for 18 h. Cells were treated with LY294002 inhibitor (Sigma, St. Louis, MO, USA) or ZnPP (R&D Systems Inc., Minneapolis, MN, USA) prior to SB treatment for 1 h. Next, 10 μL alamarBlue^®^ was added after exposure to 6-OHDA, and optical density (OD) was measured with an ELISA reader. Relative protection (percent) was calculated as 100 × ((OD of the 6-OHDA + SB-treated cells − OD of the 6-OHDA-treated cells)/(OD of the control cells − OD of the 6-OHDA-treated cells)) as described previously [72].

### 4.5. Hoechst Staining

Hoechst stain reagent (Chemicon International, Temecula, CA, USA) was used to determine the number of apoptotic cells. In brief, SH-SY5Y cells were pretreated with SB for 1 h before the 15 h treatment of 20 µM 6-OHDA. Then, 0.5 μL of Hoechst stain was added and incubated for 5 min. Cells were washed twice with wash buffer and fixed with fixative buffer. Apoptotic cells were validated by fluorescent signal with a microscope (Leica, Wetzlar, Germany, DMI-3000B), and the ratio of apoptotic cells was measured as follows: number of apoptotic cells/total cell number from five random fluorescent images [73].

### 4.6. TUNEL Staining

Apoptotic cell numbers were determined by an in Situ Cell Death Detection Kit, POD (TUNEL assay, Roche Diagnostics GmbH, Mannheim, Germany), which labeled the sites of DNA fragmentation. Cells grown on coverslips were treated for 8 h with 20 µM 6-OHDA after 1 h of incubation with SB. Cells were fixed for 1 h with a 4% paraformaldehyde solution after washing cells thrice with phosphate buffered saline (PBS). Then, a 3% H_2_O_2_ solution in methanol was used to block cells for 10 min, and then the cells were incubated with a permeabilization solution for 2 min and washed thrice with PBS. Then, the TUNEL stain mixture was added and the cells were incubated for 1 h at 37 °C. Following this, 4′,6-diamidino-2-phenylindole (DAPI) was added for 10 min, and cells were washed thrice with PBS. Coverslips were mounted with glass slides and imaged with a Leica, DM 6000 microscope. The ratio of apoptotic cell was obtained from Hoechst staining.

### 4.7. Analysis of Oxidative Stress (CellROX^®^ Staining)

The CellROX^®^ Oxidative Stress Reagent (Life Technologies, Carlsbad, CA, USA) was used to determine the levels of ROS. Cells were pretreated with SB for 1 h prior to the addition of 20 µM 6-OHDA, then they were incubated for 2 h. After further incubation at 37 °C for 30 min in the presence of 5 μM CellROX^®^ reagent, cells were washed thrice with PBS and incubated for 10 min with DAPI, and cellular ROS status was assessed using a fluorescence microscope (Leica, Wetzlar, Germany, DMI-3000B).

### 4.8. SOD Activity Assay

Cells were treated with 20 µM 6-OHDA for 24 h after 1 h pre-incubation of SB. Cells were collected and lysed with lysis buffer (5 mM β-mercaptoethanol, 0.5% Triton X-100, and 0.1 mg/mL phenylmethylsulfonyl fluoride). The mixture was then centrifuged at 4 °C, 14,000 rpm for 5 min. After the adjustment of protein concentration, samples were processed for SOD determination using the SOD activity assay kit (BioVision, Exton, PA, USA). Absorbance values were obtained on an ELISA reader (Bio Tek Instruments, Inc., Winooski, VT, USA), and the SOD inhibition level was calculated as per the manufacturer’s instructions.

### 4.9. Zebrafish Maintenance

Zebrafish (*Danio rerio*) used in this study were obtained from Tai-Kong Corporation (Taiwan), and the embryos were obtained through natural spawning. Zebrafish were fed in standard conditions (14:10 h light/dark cycle) and raised at 28.5 °C.

### 4.10. Locomotor Behavioral Test

Zebrafish larvae were pretreated with SB at 9 hpf and treated with 250 μM 6-OHDA at 2 dpf to 5 dpf in a 24-well plate. The swimming behavior of zebrafish larvae was captured by an animal behavior system utilizing automated video tracking (Singa Technology Co., Taoyuan, Taiwan, catalog no. TM-01), as described previously [40].

### 4.11. Preparation of Nuclear Extracts

Nuclear protein fraction was extracted using ProteoJET™ Cytoplasmic and Nuclear Protein Extraction Kits (Fermentas, Canada), according to the manufacturer’s instruction.

### 4.12. Western Blotting

Western blotting was performed as previously described [74]. Briefly, after adjusting for protein concentration, all samples were electrophoresed at 100 V in 10% SDSpolyacrylamide gel. Thereafter, the protein was transferred onto a poly-vinylidene fluoride (PVDF) membrane that was blocked in 5% non-fat milk in Tris Buffered Saline with Tween 20 (TTBS) for 30 min before the incubation with primary antibody overnight at 4 °C. The membrane was then incubated in secondary antibody for 1 h. Images were captured using a UVP BioChemi Imaging System.

### 4.13. Statistical Analysis

Results were presented as mean ± SEM. For Western blotting data, the intensity of each band was expressed as the relative OD divided by the average OD values from all internal controls. Data were analyzed using one-way analysis of variance followed by Dunnett’s test. The *p* values less than 0.05 were considered statistically significant.

### 4.14. Chemicals and Antibodies

6-OHDA (6-hydroxydopamine, Sigma, St. Louis, MO, USA; catalog H4381)β-Actin (a loading control; dilution, 1:1000; Sigma, St. Louis, MO, USA; catalog A5441)p-Akt (dilution, 1:1000; Cell Signaling Technology, Danvers, MA, USA; catalog 9271)Akt (dilution, 1:1000; Cell Signaling Technology, Danvers, MA, USA; catalog 9272)p-P38 (dilution, 1:1000; Cell Signaling Technology, Danvers, MA, USA; Thr180/Thr182, catalog 9211)P38 (dilution, 1:1000; Cell Signaling Technology, Danvers, MA, USA; catalog 9212)p-ERK (dilution, 1:1000; Cell Signaling Technology, Danvers, MA, USA; Thr202/204, catalog 9101)ERK (dilution, 1:1000; Cell Signaling Technology, Danvers, MA, USA; catalog 9102)Lamin b1 (dilution, 1:1000; Abcam, Biorbyt, Cambridge, UK; catalog Ab616048)Caspase-3 (dilution, 1:1000; , San Diego, CA, USA; catalog Img-144A)HO-1 (dilution, 1:1000; Cell Signaling Technology, Danvers, MA, USA; catalog 5061)Nrf2 (dilution, 1:1000; Abcam, Biorbyt, Cambridge, UK; catalog Ab31163)TH (tyrosine hydroxylase; dilution 1:1000; Millipore, Billerica, MA, USA; catalog Mab318)LY294002 (2-(4-Morpholinyl)-8-phenyl-1(4H)-benzopyran-4-one hydrochloride, Sigma, St. Louis, MO, USA; catalog L9908)ZnPP (Sigma, St. Louis, MO, USA; catalog 691550-M)

## 5. Conclusions

Neuroprotective mechanism of action of SB in neuronal cell apoptosis: Treatment with 6-OHDA increases intracellular ROS and subsequent antioxidant response element (ARE) activity. Enhanced ARE activity triggers the binding of transcription factor Nrf2 to the ARE domain in the nucleus, signaling the production of HO-1 and SOD-1. Treatment with 6-OHDA attenuates the expression of p-Akt and p-ERK and enhances phosphorylation of P38, thereby activating the downstream caspase-3 cascade. Pretreatment with SB reverses the 6-OHDA-induced downregulation of the PI3K/Akt signaling pathway and further enhances the translocation of Nrf2 to promote downstream protein translation of HO-1 and SOD-1. Also, SB inhibits the cleavage of caspase-3 protein through upregulation of p-Akt and p-ERK and attenuation of p-P38. Through these anti-apoptotic and anti-oxidative effects in SH-SY5Y cells and zebrafish, SB protects against cellular damage and rescues zebrafish from the 6-OHDA-induced locomotor deficiency.

## Figures and Tables

**Figure 1 marinedrugs-17-00315-f001:**
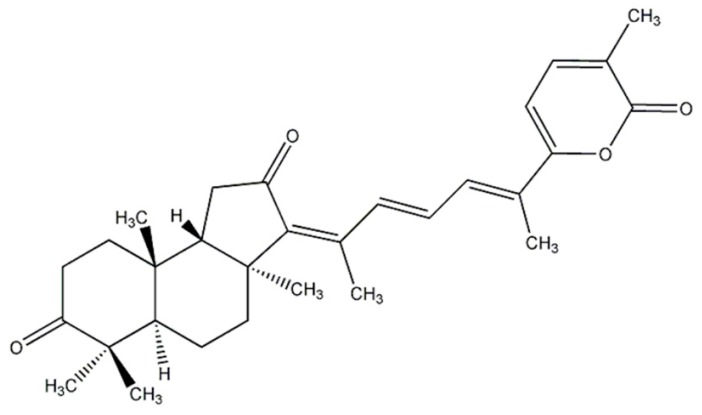
Chemical structure of stellettin B (SB).

**Figure 2 marinedrugs-17-00315-f002:**
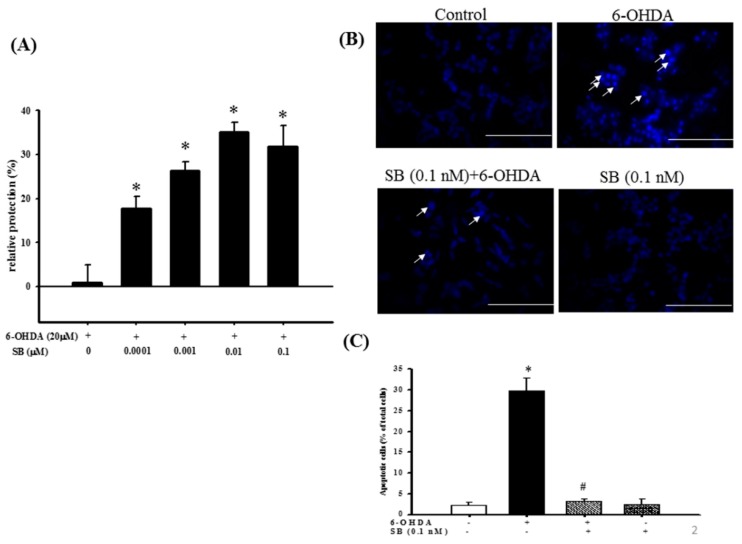
Cytoprotective effect of SB against 6-OHDA damage in SH-SY5Y cells: (**A**) SH-SY5Y cells were pretreated with 0.1, 1, 10, or 100 nM SB for 1 h and then challenged with 20 μM 6-OHDA for 16 h. Apoptosis in the 6-OHDA-treated group was normalized to 0%. Data are presented as mean ± SEM, and each value represents the mean of three replicates and six samples. * significantly different from the 6-OHDA group; (**B**) SH-SY5Y cells were pretreated with 0.1 nM SB for 1 h and then challenged with 20 μM 6-OHDA for 8 h. Hoechst 33342 stainings of the control, 6-OHDA, 6-OHDA plus SB, and SB alone groups are shown. The white arrows indicate the locations of chromatin condensation (scale bar = 100 μM); (**C**) Quantification of cytotoxicity in each group. Data are presented as mean ± SEM, and each value represents the mean of three replicates and three samples. *significantly different from the control group; # significantly different from the 6-OHDA group. *p* < 0.05.

**Figure 3 marinedrugs-17-00315-f003:**
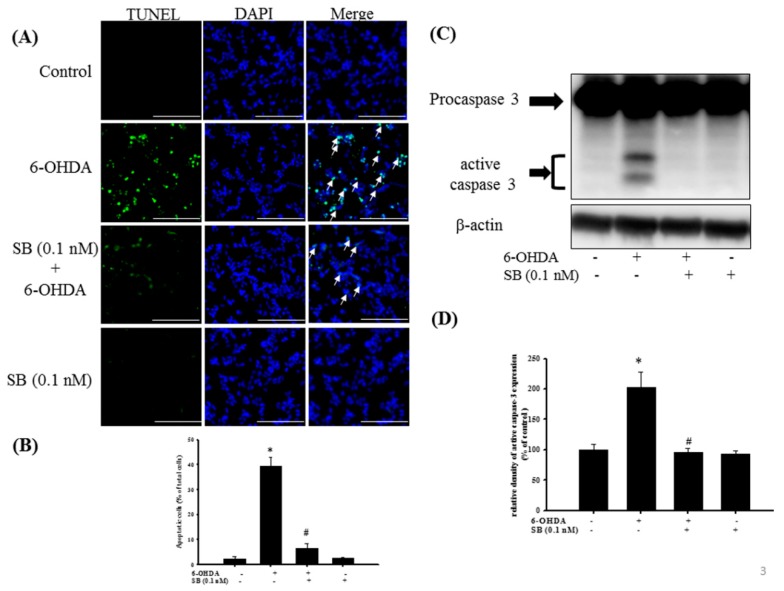
The anti-apoptotic effect of SB on 6-OHDA-induced neurotoxicity in SH-SY5Y cells: SH-SY5Y cells were pretreated with 0.1 nM SB for 1 h and then challenged with 20 μM 6-OHDA for 8 h in the control, 6-OHDA, 6-OHDA plus SB, and SB alone treatment groups. (**A**) TUNEL staining. White arrows indicate apoptotic cells (scale bar = 100 μM); (**B**) Quantification of apoptotic cells in each treatment group; (**C**) Western blotting showing induction of cleaved caspase-3 protein; (**D**) Quantification of relative density of cleaved caspase-3 protein from Western blotting. Data are presented as mean ± SEM, and each value represents the mean of three replicates and three samples. *significantly different from the control group; # significantly different from the 6-OHDA group.

**Figure 4 marinedrugs-17-00315-f004:**
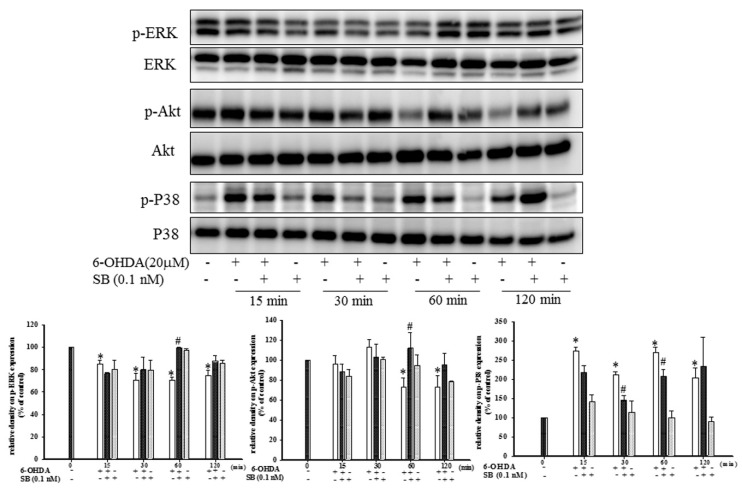
Effect of SB on 6-OHDA-induced downregulation of phospho-extracellular signal-regulated kinases (p-ERK) and phospho-protein kinase B (p-Akt) and the increase in phospho-P38 in SH-SY5Y cells. SH-SY5Y cells were pretreated with 0.1 nM SB for 1 h and then challenged with 20 μM 6-OHDA for 15, 30, 60, or 120 min. Western blottings for p-ERK, p-Akt, and p-P38 of the control, 6-OHDA plus SB, and SB alone treatment groups are shown. Total ERK, Akt, and P38 were determined as an internal control of each band. Data are presented as mean ± SEM, and each value represents the mean of three replicates and three samples. *significant compared with the control group; # significant compared with the 6-OHDA group.

**Figure 5 marinedrugs-17-00315-f005:**
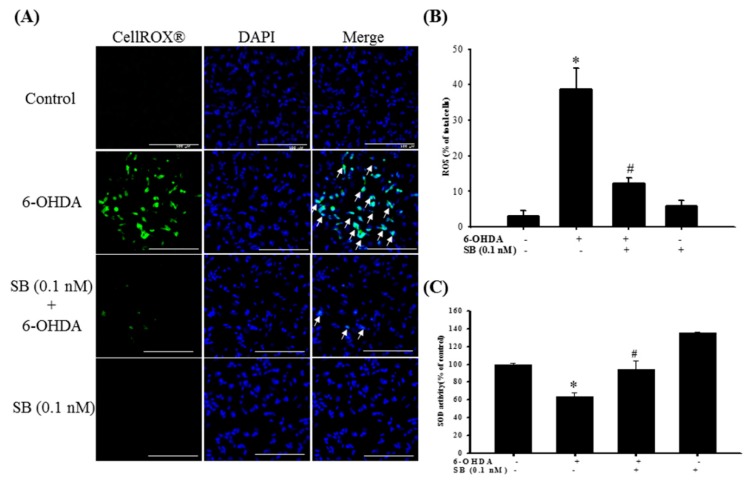
Effect of SB on 6-OHDA-induced upregulation of intracellular reactive oxygen species (ROS) and downregulation of superoxide dismutase (SOD) activity. SH-SY5Y cells were pretreated with 0.1 nM SB for 1 h and then challenged with 20 μM 6-OHDA for 2 h. (**A**) CellROX^®^ of the control, 6-OHDA, 6-OHDA plus SB, and SB treatment groups. White arrows indicate the location of ROS-positive cells (scale bar = 100 μM). (**B**) Quantification of the frequency of ROS-positive cells in each group. (**C**) SOD levels of the control, 6-OHDA, 6-OHDA plus SB, and SB alone treatment groups. Data are presented as mean ± SEM, and each value represents the mean of three replicates and three samples. *significant compared with the control group; # significant compared with the 6-OHDA group.

**Figure 6 marinedrugs-17-00315-f006:**
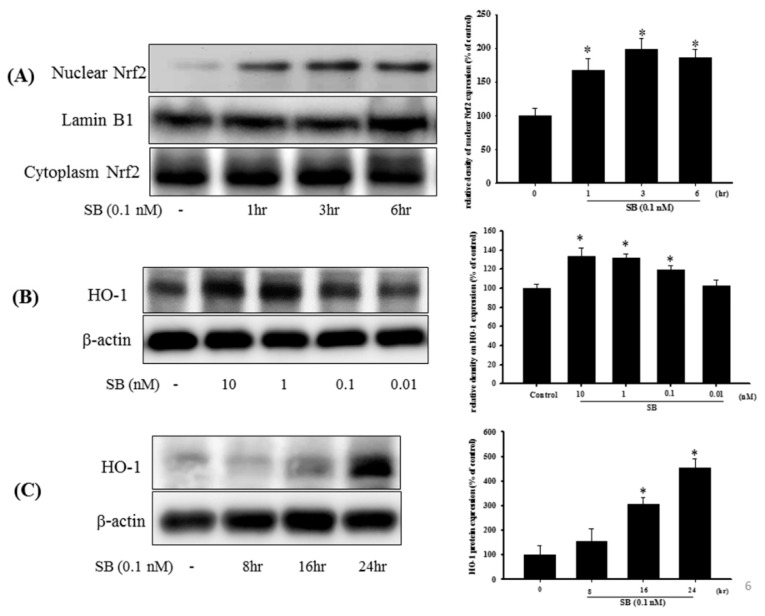
Effect of SB on nuclear factor erythroid-2-related factor (Nrf2) and heme-oxygenase 1 (HO-1) expression in SH-SY5Y cells. (**A**) SH-SY5Y cells were treated with 0.1 nM SB for 1, 3, or 6 h, and nuclear proteins were isolated using a Nuclear Extraction Kit followed by Western blotting for the Nrf2 protein using lamin B1 as an internal control; (**B**) Western blotting for HO-1 protein after treatment with SB at 0.01, 0.1, 1, or 10 nM for 16 h using β-actin as an internal control; (**C**) SH-SY5Y cells were treated with 0.1 nM SB for 8, 16, or 24 h followed by Western blotting for HO-1 protein using β-actin as an internal control. Quantification of blots is shown in the graphs at the right, and data are presented as mean ± SEM with each value representing the mean of three replicates and three samples. * Significant compared with the control group.

**Figure 7 marinedrugs-17-00315-f007:**
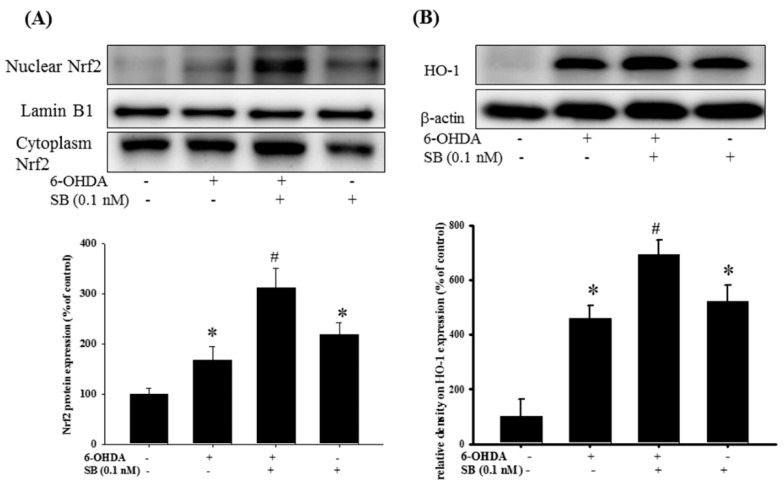
Effect of SB on nuclear factor erythroid-2-related factor (Nrf2) and heme-oxygenase 1 (HO-1) expression in 6-OHDA-treated SH-SY5Y cells: SH-SY5Y cells were pretreated with 0.1 nM SB for 1 h and then challenged with 20 μM 6-OHDA for 3 h. Nuclear protein was isolated with a Nuclear Extraction Kit; (**A**) Western blotting for Nrf2 protein in the control, 6-OHDA, 6-OHDA plus SB, and SB alone treatment groups. Lamin B1 was used as a nuclear internal control; (**B**) Western blotting for HO-1 using β-actin as an internal control. Quantification of blots is shown in the graphs below, and data are presented as mean ± SEM with each value representing the mean of three replicates and three samples. * Significant compared with the control group; # significant compared with the 6-OHDA group.

**Figure 8 marinedrugs-17-00315-f008:**
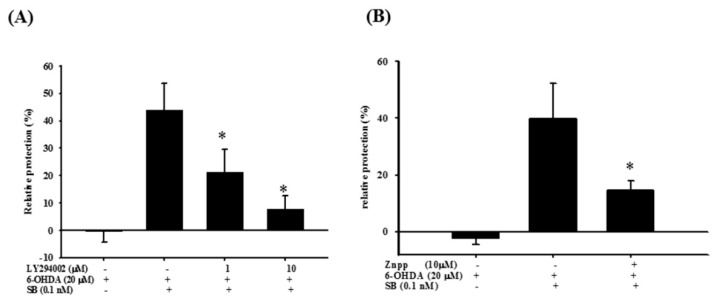
Effect of LY294002 and zinc protoporphyrin (ZnPP) on the neuroprotective effect of SB in 6-OHDA-treated SH-SY5Y cells. (**A**) SH-SY5Y cells were pretreated with 1 or 10 μM LY294002 for 1 h, then with 0.1 nM SB for another 1 h, followed by 20 µM 6-OHDA for 18 h. Relative protection in the 6-OHDA, 6-OHDA plus SB, 6-OHDA plus SB plus 1 μM LY294002, and 6-OHDA plus SB plus 10 μM LY294002 groups is shown; (**B**) SH-SY5Y cells were pretreated with 10 μM ZnPP for 1 h, then with 0.1 nM SB for another 1 h, followed by 20 µM 6-OHDA for 18 h. Relative protection in 6-OHDA, 6-OHDA plus SB, and 6-OHDA plus SB plus 10 μM ZnPP groups is shown. Data are presented as mean ± SEM, and each value represents the mean of three replicates and three samples. * Significant compared with the 6-OHDA group.

**Figure 9 marinedrugs-17-00315-f009:**
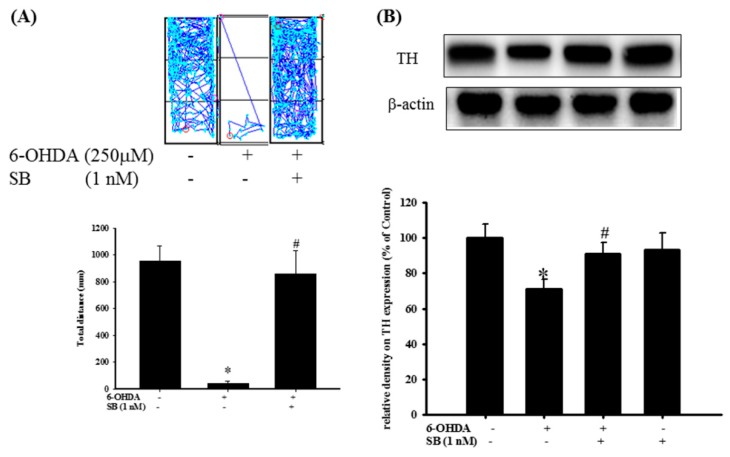
Effect of SB on 6-OHDA-induced locomotor deficiency and tyrosine hydroxylase (TH) expression in zebrafish. (**A**) Zebrafish larvae were pretreated with 1 nM SB from 9 h post-fertilization (hpf) to 5 days post-fertilization (dpf) and then challenged with 250 µM 6-OHDA from 2 to 5 dpf. The upper panel shows representative data from one experiment, whereas the lower panel shows average results. Data are presented as mean ± SEM, and each value represents the mean 16 samples. * Significant compared with the control group; #compared with the 6-OHDA group; (**B**) Zebrafish larvae were pretreated with 1 nM SB from 9 hpf to 4 dpf and then challenged with 250 µM 6-OHDA from 2 to 4 dpf; Western blotting for TH of the control, 6-OHDA, 6-OHDA plus SB, and SB alone in 4 dpf treatment groups is shown using β-actin as an internal control. Data are presented as mean ± SEM, and each value represents the mean of three replicates and three samples. Each band contained pooled material from 20 fish. * Significant compared with the control group; # significant compared with the 6-OHDA group.

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
