# Peer review of "In Vitro and In Vivo Neuroprotective Effects of Stellettin B Through Anti-Apoptosis and the Nrf2/HO-1 Pathway"

_marinedrugs, 2019, doi:10.3390/md17060315_

Round 1
Reviewer 1 Report
Title: In vitro and in vivo neuroprotective effect of Stellettin B through anti-apoptosis and the Nrf2/HO-1 Pathway.
The manuscript represents an interesting, well-constructed research paper with well-planned studies. It describes the protective role of Stellettin B (SB) in vitro using the neuroblastoma cell line SH-SY5Y, a widely accepted neuronal cell model and in vivo with zebrafish.
The authors provide supporting evidence that SB conferred neuroprotection against 6-hydroxydopamine (6-OHDA) toxicity in SH-SY5Y via reduction of pro-apoptotic and pro-oxidant molecules. Furthermore, the authors investigated whether the therapeutic effects of SB could be translated in vivo in zebrafish Parkinson’s disease model. They report that SB reversed the swimming behavior and increased the number of dopaminergic neurons expressing tyrosine hydroxylase in a 6-OHDA zebrafish model. Overall the evidence supporting the therapeutic effect of SB in 6-OHDA model of PD is convincing. The manuscript is well written and has no major flaws. However, I would recommend the following minor modifications:
Line 445: 5. Conclusions: A written paragraph should be included in this section in addition to the figure. The authors should consider using the description in figure as the conclusion while adding a short legend for the figure.
Line 268: Reference 41 should be deleted. This reference describes the procedure for “An improved stain for Heinz bodies” and has nothing to do with treatment options for Parkinson’s disease.
Author Response
Response to Reviewer 1 Comments
Line 445: 5. Conclusions: A written paragraph should be included in this section in addition to the figure. The authors should consider using the description in figure as the conclusion while adding a short legend for the figure.
Our reply: Thanks for your suggestion, conclusive figure with figure legend as follow has been added to the conclusion section:
“Neuroprotective mechanism of action of SB in neuronal cell apoptosis. Treatment with 6-OHDA increases intracellular ROS and subsequent antioxidant response element (ARE) activity. Enhanced ARE activity triggers the binding of transcription factor Nrf2 to the ARE domain in nucleus, signaling the production of HO-1 and SOD-1. Treatment with 6-OHDA attenuates the expression of p-Akt and p-ERK and enhances phosphorylation of p38, thereby activating the downstream caspase-3 cascade. Pretreatment with SB reverses 6-OHDA-induced downregulation of PI3K/Akt signaling pathway and further enhances the translocation of Nrf2 to promote downstream protein translation of HO-1 and SOD-1. Also, SB inhibits the cleavage of caspase-3 protein through upregulation of p-Akt and p-ERK and attenuation of p-P38. Through these anti-apoptotic and anti-oxidative effects in SH-SY5Y cell and zebrafish, SB protects against cellular damage and rescues zebrafish from the 6-OHDA-induced locomotor deficiency.” in line 452-463
Line 268: Reference 41 should be deleted. This reference describes the procedure for “An improved stain for Heinz bodies” and has nothing to do with treatment options for Parkinson’s disease.
Our reply: Thanks for your careful review, we have deleted the reference as per your suggestion.
Reviewer 2 Report
1- There are several linguistic mistakes, lake of presenting a proper summery of the work in the abstract. Please re- write the abstract as its so vague.
2- The authors talked about PI3K/Akt, MAPK/p38 in line 65-69 in the introduction, How this related to their work? this is not clear. This need to be added to the abstract if the work included measuring any effects on these pathways.
3- The introduction seems vague, and missing linking between idea. What is the gap that they need to highlight here?
4- Lines 313-316 in the discussion does not make any sense, please exclude it.
5- Lines 317-324 are very vague, What the authors trying to say here? and how this linked to the next following lines? need to be re-phrased.
6- Lines 333 - 338 there is no link between the ideas. again its not clear what the authors try to discuss or show here.
7- Line 350 the preparation of SB, authors must show the extraction methods that been used and more details about the yield product. Or at least cite the original work that extracted the product from marine sponge.
Author Response
Response to Reviewer 2 Comments
Comments and Suggestions for Authors
1- There are several linguistic mistakes, lake of presenting a proper summery of the work in the abstract. Please re- write the abstract as its so vague.
Our reply: Thanks for the suggestion, we have rewritten the abstract as follow:
“Pharmaceutical agents for halting the progression of Parkinson's disease (PD) are lacking. The current available medications only relieve clinical symptoms and may cause severe side effects. Therefore, there is an urgent need for novel drug candidates for PD. In this study, we demonstrated the neuroprotective activity of stellettin B (SB), a compound isolated from marine sponge. We showed that SB could significantly protect SH-SY5Y cells against 6-OHDA-induced cellular damage by inhibiting cell apoptosis and oxidative stress through PI3K/Akt, MAPK, caspase cascade and Nrf2/HO-1 cascade modulation respectively. In addition, in vivo study showed that SB reversed 6-OHDA-induced locomotor deficit in a zebrafish model of PD. The potential for developing SB as a candidate drug for PD treatment is discussed.”
2- The authors talked about PI3K/Akt, MAPK/p38 in line 65-69 in the introduction, How this related to their work? this is not clear. This need to be added to the abstract if the work included measuring any effects on these pathways.
Our reply: Thanks for the suggestion, it is widely known that dopaminergic cell survival or anti-apoptosis involves activation of PI3K-Akt and inhibition of MAPK/p38 [1, 2]. In order to solve the current medical dilemma, we intend to examine if this marine-derived compound could serve as a novel PD drug candidate via PI3K/Akt or MAPK/p38 pathway modulation. Hence, we have reedited the line 67-72 as follow:
“The roles of PI3K/Akt pathway in regulating neuronal cell survival, proliferation and differentiation have been widely investigated in PD [3, 4]. On the other hand, MAPK/p38 modulates stress responses and apoptosis in dopaminergic neurons [5]. Moreover, previous studies indicate that activation of PI3K/Akt and inhibition of MAPK/p38 block apoptosis and promote dopaminergic cell survival [1, 2]. Thus, the modulatory effect of candidate compounds on PI3K/Akt or MAPK/P38 pathway may be import in exerting neuroprotective activity. “
Also, we have corrected the abstract as follow:
“Pharmaceutical agents for halting the progression of Parkinson's disease (PD) are lacking. The current available medications only relieve clinical symptoms and may cause severe side effects. Therefore, there is an urgent need for novel drug candidates for PD. In this study, we demonstrated the neuroprotective activity of stellettin B (SB), a compound isolated from marine sponge. We showed that SB could significantly protect SH-SY5Y cells against 6-OHDA-induced cellular damage by inhibiting cell apoptosis and oxidative stress through modulating PI3K/Akt, MAPK, caspase cascade and Nrf2/HO-1 cascade respectively. Also, in vivo study showed that SB reversed 6-OHDA-induced locomotor deficit in a zebrafish model of PD. The potential for developing SB as a candidate drug for PD treatment is discussed.”
1. Bohush, A.; Niewiadomska, G.; Filipek, A., Role of Mitogen Activated Protein Kinase Signaling in Parkinson's Disease. Int J Mol Sci 2018, 19, (10).
2. Jha, S. K.; Jha, N. K.; Kar, R.; Ambasta, R. K.; Kumar, P., p38 MAPK and PI3K/AKT Signalling Cascades inParkinson's Disease. Int J Mol Cell Med 2015, 4, (2), 67-86.
3. Zhang, Y.; He, Q.; Dong, J.; Jia, Z.; Hao, F.; Shan, C., Effects of epigallocatechin-3-gallate on proliferation and differentiation of mouse cochlear neural stem cells: Involvement of PI3K/Akt signaling pathway. Eur J Pharm Sci 2016, 88, 267-73.
4. Yu, J. S.; Cui, W., Proliferation, survival and metabolism: the role of PI3K/AKT/mTOR signalling in pluripotency and cell fate determination. Development 2016, 143, (17), 3050-60.
5. Sun, Y.; Liu, W. Z.; Liu, T.; Feng, X.; Yang, N.; Zhou, H. F., Signaling pathway of MAPK/ERK in cell proliferation, differentiation, migration, senescence and apoptosis. J Recept Signal Transduct Res 2015, 35, (6), 600-4.
3- The introduction seems vague, and missing linking between idea. What is the gap that they need to highlight here?
Our reply: Thanks for the valuable suggestion, we have added appropriate sentences in the introduction section to link the ideas in revised manuscript.
“Apoptosis were affected by numerous pathways, included PI3K/Akt, MAPK/p38 and caspase cascade.” in line 66-67
“Ocean serves as a potential biological resource for novel PD drug discovery due to its biological diversity. Also, marine-derived drugs may solve the current drug shortage issues [1].” in line 73-75
1. Gogineni, V.; Hamann, M. T., Marine natural product peptides with therapeutic potential: Chemistry, biosynthesis, and pharmacology. Biochim Biophys Acta Gen Subj 2018, 1862, (1), 81-196.
4- Lines 313-316 in the discussion does not make any sense, please exclude it.
Our reply: Thanks for the careful review, we have deleted this paragraph as per your suggestion.
5- Lines 317-324 are very vague, What the authors trying to say here? and how this linked to the next following lines? need to be re-phrased.
Our reply: Thanks for the valuable suggestion, we would like introduce some previous research which evaluated the neuroprotective activity in a zebrafish PD model. As your suggestion, we have added the following sentences to clarify the idea.
“Some research also showed a resemble result in zebrafish model as our study. Xu et al showed that Loganin, one of the best-known iridoid glycosides, exerts a neuroprotective effect in an MPTP-mouse model of PD through anti-inflammatory, autophagy attenuation, and anti-apoptotic activities [1].” in line 318-321
1. Xu, Y. D.; Cui, C.; Sun, M. F.; Zhu, Y. L.; Chu, M.; Shi, Y. W.; Lin, S. L.; Yang, X. S.; Shen, Y. Q., Neuroprotective Effects of Loganin on MPTP-Induced Parkinson's Disease Mice: Neurochemistry, Glial Reaction and Autophagy Studies. J Cell Biochem 2017, 118, (10), 3495-3510.
6- Lines 333 - 338 there is no link between the ideas. again its not clear what the authors try to discuss or show here.
Our reply: Thanks for the careful review, we would like to discuss that SB shows anti-tumor activity in previous research, [1, 2] even in brain cancer[3]. We discovered the protective effect of SB on neuron in our present study. To further clarify our idea, we have corrected line 333-338 as follow:
“To expand the range of drugs available to tackle PD, we searched and identified marine sponge-derived SB, which was hypothesized not to possess neuroprotective property based on its previous literature. Earlier studies have focused on SB's antitumor effect in human non-small cell lung cancer and glioblastoma [1-3]. To our excitement, we demonstrated its neuroprotective activity in a PD model albeit at a lower dosage.” in line line 333-337
1. Tang, S. A.; Zhou, Q.; Guo, W. Z.; Qiu, Y.; Wang, R.; Jin, M.; Zhang, W.; Li, K.; Yamori, T.; Dan, S.; Kong, D., In vitro antitumor activity of stellettin B, a triterpene from marine sponge Jaspis stellifera, on human glioblastoma cancer SF295 cells. Mar Drugs 2014, 12, (7), 4200-13.
2. Wang, R.; Zhang, Q.; Peng, X.; Zhou, C.; Zhong, Y.; Chen, X.; Qiu, Y.; Jin, M.; Gong, M.; Kong, D., Stellettin B Induces G1 Arrest, Apoptosis and Autophagy in Human Non-small Cell Lung Cancer A549 Cells via Blocking PI3K/Akt/mTOR Pathway. Sci Rep 2016, 6, 27071.
3. Cheng, S. Y.; Chen, N. F.; Lin, P. Y.; Su, J. H.; Chen, B. H.; Kuo, H. M.; Sung, C. S.; Sung, P. J.; Wen, Z. H.; Chen, W. F., Anti-Invasion and Antiangiogenic Effects of Stellettin B through Inhibition of the Akt/Girdin Signaling Pathway and VEGF in Glioblastoma Cells. Cancers (Basel) 2019, 11, (2).
7- Line 350 the preparation of SB, authors must show the extraction methods that been used and more details about the yield product. Or at least cite the original work that extracted the product from marine sponge.
Our reply: Thanks for noticing us, we have added the following sentences.
“Stellettin B was obtained from the sponge, Jaspis stellifera. The sponge was thoroughly extracted with methanol. Then, the methanol extract was subjected to further partition between H2O and CH2Cl2. The CH2Cl2 layer was separated over a silica gel by column chromatography to obtain stellettin B.” in line 355-358.